# RoCo: Robust Collaborative Perception By Iterative Object Matching and Pose Adjustment

## ABSTRACT

Collaborative autonomous driving with multiple vehicles usually requires the data fusion from multiple modalities. To ensure effective fusion, the data from each individual modality shall maintain a reasonably high quality. However, in collaborative perception, the quality of object detection based on a modality is highly sensitive to the relative pose errors among the agents. It leads to feature misalignment and significantly reduces collaborative performance. To address this issue, we propose RoCo, a novel unsupervised framework to conduct iterative object matching and agent pose adjustment. To the best of our knowledge, our work is the first to model the pose correction problem in collaborative perception as an object matching task, which reliably associates common objects detected by different agents; On top of this, we propose a graph optimization process to adjust the agent poses by minimizing the alignment errors of the associated objects, and the object matching is re-done based on the adjusted agent poses. This process is iteratively repeated until convergence. Experimental study on both simulated and real-world datasets demonstrates that the proposed framework RoCo consistently outperforms existing relevant methods in terms of the collaborative object detection performance, and exhibits highly desired robustness when the pose information of agents is with high-level noise. Ablation studies are also provide to show the impact of its key parameters and components. The code will be released.

## CCS CONCEPTS

• **Computer system organization** → **Autonomous driving**; • **Computing methodologies** → *Collaborative detection*.

## KEYWORDS

Collaborative Perception, 3D object detection, Point cloud, Pose error, Graph Matching

## 1 INTRODUCTION

Collaborative perception can significantly enhance perception performance by sharing information from different sensors among agents. It can overcome the inherent limitations of single-agent-based perception, such as invisibility caused by occlusion or long-range issues. Recent research [13, 15, 18, 40, 43] has spurred the widespread attention in the fields such as autonomous driving,

**Unpublished working draft. Not for distribution.**

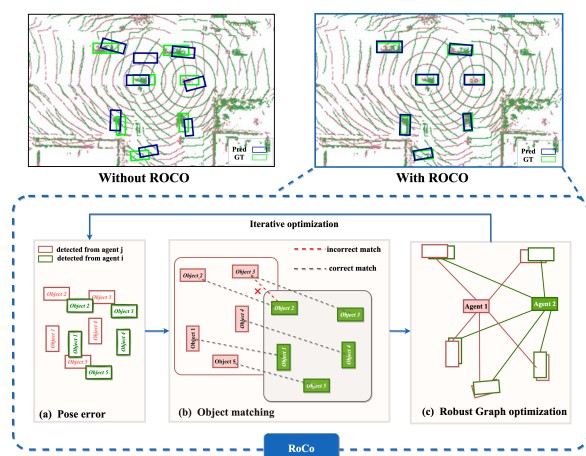

**Figure 1: Illustration of robust collaborative perception system and the result with or without the proposed RoCo.**

metaverse, multi-robot systems, and multimedia application systems [1, 9, 16, 19, 22, 38, 41, 44]. As an emerging topic, collaborative 3D object detection is of significance but also faces many challenges, including model-agnostic and task-agnostic formulation [23], communication bandwidth constraints [15, 35], time delay [36], adversarial attacks [21, 37], and multimodal fusion [24, 26].

In collaborative 3D object detection, regardless of the modality of the employed sensors, the detection module of each agent transmits the obtained object detection features to the Ego agent. The Ego agent then gathers the features of the same object and then carries out further information fusion. Therefore, the object detection features received from other agents need to be accurately aligned in the Ego agent for further processing. However, in practical applications, the pose estimated for each agent contains errors, which can in turn cause misalignment of the positions of the same objects across agents. For example, as illustrated in Figure 1(a), this misalignment may result in the incorrect fusion of *Object 3* detected by agent *i* (red box) with *Object 2* detected by agent *j* (green box)). This issue becomes more pronounced when the pose error is large or high traffic is presented. This paper will focus on addressing the object alignment problem between multiple agents due to pose errors in a given modality (LiDAR).

Previous works have proposed some methods to improve the robustness of pose estimation [11, 32, 35, 40, 45]. For example, V2X-ViT [40] adopts multiscale window attention to capture features at different ranges, while V2VNet [35] combines pose regression, global consistency, and attention aggregation modules to correct relative poses. CoAlign [25] introduces a proxy-object pose graph to enhance the pose consistency between collaborative agents without the need for supervision of real poses. The above methods work well when there is minimal variation in poses and a limited number

of vehicles. However, in crowded and complex scenarios, as the number of objects and the magnitude of pose errors increase, the deviation in object positions will grow, making it increasingly difficult to correctly match objects at the same location. This situation ultimately reduces the accuracy of 3D object detection.

To address these limitations, we propose a novel iterative object matching and pose adjustment framework called RoCo, which can handle the issue of matching the object detection information from multiple vehicles in noisy scenarios. RoCo is primarily divided into two parts: object matching and robust graph optimization. Object matching involves correctly establishing the matching relationships between multiple detected objects using the distance and neighborhood structural consistency of local graphs. Robust graph optimization builds a pose optimization graph for the whole scene based on the above matching relationships, and iteratively adjusts the poses of agents according to global observation consistency, so as to effectively filters out incorrect matches. RoCo has three main advantages: i) it is an unsupervised method, requiring no ground-truth pose information for the agents or objects and it can adapt to various levels of pose errors. ii) Implementing RoCo does not require retraining or fine-tuning any network models and can be integrated into any 3D object detection based collaborative perception framework. iii) Even in crowded and noisy environments, RoCo can accurately align the detections of the same object obtained by different agents. Unlike the existing methods that adjust object detection features, this proposed method is based on an object-matching-guided pose adjustment which can fundamentally prevent the introduction of additional noise into the features.

We conduct extensive experiments on both simulation and real-world datasets, including V2XSet[40] and DAIR-V2X[44]. Results show that RoCo consistently achieves the best performance in the task of collaborative 3D object detection with the presence of pose errors. In summary, the main contribution of this work are:

- We propose RoCo, a novel robust multi-agent collaborative LiDAR-based 3D object detection framework that addresses the matching errors and pose inaccuracies between agents and objects. To the best of our knowledge, RoCo is the first to model the pose correction problem in collaborative perception as an object matching task.
- The proposed RoCo establishes matching relationships between targets based on distance and neighborhood structural consistency using a graph matching approach. On top of this, RoCo iteratively adjusts agent poses based on global observation consistency, effectively filtering out incorrect object matches.
- Extensive experiments have shown that RoCo achieves more accurate and robust 3D object detection performance even in the scenarios with vehicle congestion and significant noise.

## 2 RELATED WORK

### 2.1 Collaborative Perception

Collaborative perception is a promising mechanism in multi-agent systems that can overcome the limits of single-agent perception. To support research in this field, various strategies have been devised to address practical challenges[39][34][24]. To reduce communication bandwidth, Where2comm[15] selects spatially sparse but perceptually significant regions by using a spatial confidence map. V2X-ViT[40] includes a multi-agent attention module that aggregates information from various agents, improving detection performance. To handle communication delays, SyncNet[27] uses historical multi-frame information to compensate for current information, employing feature attention co-symbiotic estimation and time modulation techniques. CoBEVFlow[36] creates a synchrony-robust collaborative system for Bird's Eye View (BEV) flow that aligns asynchronous collaboration messages sent by various agents using motion compensation. HEAL[24] introduces a pioneering and adaptable collaborative perception framework designed to seamlessly integrate continually emerging heterogeneous agent types into collaborative perception tasks. However, in crowded and noisy road scenarios, information often becomes inconsistent and misaligned, leading to mismatches between objects detected by different agents and resulting in performance degradation. Therefore, in this work, we specifically considered the robustness to pose errors.

### 2.2 Noisy Pose Issue

Due to performance discrepancies between hardware acquisition devices and model estimations, the poses of agents are susceptible to interference, leading to misalignment and inconsistencies during fusion processes. Therefore, many methods attempt to design robust networks to correct error influences. The first category of methods primarily involves proposing robust network architectures and introducing specific modules to learn the influence of poses, such as V2VNet[35], MASH[12], V2X-ViT[40] and FeaCo[13]. However, these methods require additional pose supervision signals. The second category is to use the coarse observations as prior knowledge to adjust the poses of objects[25, 50]. CoAlign[25] introduced a method for modeling and optimizing agent-object pose graphs to enhance the consistency of relative poses, thereby improving model robustness. Nevertheless, in circumstances with crowded roadways, this approach's use of clustering to create connections between objects could result in mistakes and fail pose graph optimisation. In contrast, our proposed RoCo models the reduction in pose errors as a matching and optimization problem for objects, accurately identifying relationships between objects detected by multiple agents. Through iterative pose optimization based on the correct matching graph, RoCo improves the accuracy of 3D detection.

### 2.3 Object Matching

SLAM[3, 4, 29, 30, 50] (Simultaneous Localization and Mapping) is a crucial research approach in scene understanding, primarily used by autonomous robots (such as robots and autonomous vehicles) to estimate their positions and build maps of the environment simultaneously during motion, without prior environmental information. Object matching in SLAM refers to associating detection results of the current frame with known objects in the scene. Most methods use metrics such as distances between different objects or Intersection over Union (IoU), followed by matching using the Hungarian algorithm[6, 31, 33, 47, 49], which is classical and high-performing but impractical for large-scale deployment in real environments. Therefore, researchers have explored many methods to facilitate its application[2][5][7]. *To our knowledge, we are the first to model the pose correction problem of multiple agents in collaborative perception*

 

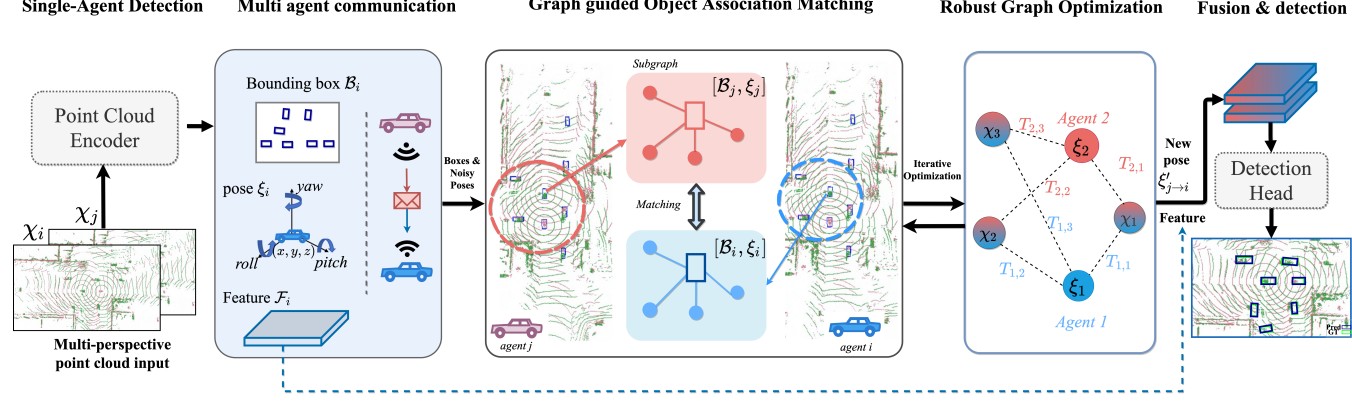

**Figure 2: Overview of RoCo system. The object bounding boxes and poses are transmitted as messages to other agents to achieve object matching and robust graph optimization, resulting in corrected matching and poses. Features are transformed based on the corrected poses in the ego coordinate system and fused across all agents.**

*as an object matching and optimization task.* This work marks the first application of SLAM-based object matching in collaborative perception, expanding its scope of application, and investigation into the impact of object matching on multi-agent collaborative perception systems in challenging environments.

## 3 COLLABORATIVE PERCEPTION AND THE ISSUE OF POSE ERROR

Following the literature of collaborative perception [13, 25, 36], the commonly used model can be described as below.

It is assumed that there are $N$ agents (i.e., collaborators) in the scene. Let $\mathcal{X}_i$ and $O_i$ be the raw observation and the perception output of Ego agent $i$. $\mathcal{M}_{j \to i}$ denotes the collaboration message sent from agent $j$ to agent $i$. The goal of collaborative perception is to enhance the 3D object detection capability of the ego agent through cooperation. This process can be formally expressed as:

$$F_i = \Phi_{Enc}(\mathcal{X}_i), \quad i = 1, \cdots, N, \tag{1a}$$

$$\mathcal{M}_{j \to i} = \Phi_{Proj}(\xi_i, (F_j, \xi_j)), \quad j = 1, \cdots, N; j \neq i \tag{1b}$$

$$F_i' = \Phi_{Agg}(F_i, \{\mathcal{M}_{j \to i}\}_{j=1,2,\ldots,N; j \neq i}), \tag{1c}$$

$$O_i = \Phi_{Dec}(F_i'). \tag{1d}$$

First, for each agent, Step (1a) extracts feature $F_i$ from the raw observation $\mathcal{X}_i$ via an encode network $\Phi_{Enc}(\cdot)$. After that, each agent $j$ projects, via an projection module $\Phi_{Proj}(\cdot)$, its feature $F_j$ to the agent $i$'s coordinate system based on $\xi_i$ and $\xi_j$ which represent the poses of the two agents, respectively. The projected feature is then sent to agent $i$ as a message $\mathcal{M}_{j \to i}$, and this completes Step (1b). After receiving all the $(N-1)$ messages, agent $i$ will fuse its feature $F_i$ with these messages via a fusion network $\Phi_{Agg}(\cdot)$, producing a fused feature $F_i'$ as in Step (1c). Finally, Step (1d) uses a decode network $\Phi_{Dec}(\cdot)$ to convert $F_i'$ to the final perception output $O_i$.

As seen above, when the pose of agent $i$ is assumed to be given, the projection in Step (1b) will critically rely on the accuracy of the pose of agent $j$, $\xi_j$. However, the pose estimated by each agent cannot be perfect in practice and often come with noise, i.e., $\{\xi_j, j =$

$1, \cdots, N\}$ have errors. This kind of error adversely affects the accuracy of the projected feature to be sent via the message $\mathcal{M}_{j \to i}$. This leads to the misalignment of features when the message is processed by agent $i$ and finally hurts the performance of 3D object detection.

To correct agent poses, existing work [25] innovatively uses a clustering based method to determine association among the bounding boxes detected by different agents. After finding object associations, the agent poses are adjusted by graph optimization accordingly. Nevertheless, when the error of agent pose becomes significant or when objects become close in crowded scenes, such a clustering based matching method will be prone to producing large matching errors. The matching errors will in turn lead to large pose adjusting errors, causing poor collaborative perception.

## 4 OUR PROPOSED METHOD

To improve this situation, this paper proposes an unsupervised, iterative object matching and pose adjusting framework, called **RoCo**. It can maintain accurate matching relationships even in scenarios with high pose errors and traffic congestion, thereby enhancing object detection performance.

Our solution consists of two key ideas. Firstly, finding reliable matching relationship between two objects depends not only on their spatial similarity but also on the similarity of the object configuration in the neighborhood of the two objects. Secondly, we take an iterative approach to conduct object matching and pose adjustment. That is, after using the current matching result to adjust the agent poses, object matching will be conducted again upon the updated poses. This process repeats until convergence, that is, the object matching does not change. This approach is able to improve the matching accuracy and in turn lead to better adjustment for the agent poses.

It is worth noting that the accuracy of object detection from each individual agent also affects the quality of object matching. In this work, we assume that every agent can independently and accurately detect objects within its own range. This allows us to focus on the issue of feature misalignment caused by pose errors.

 

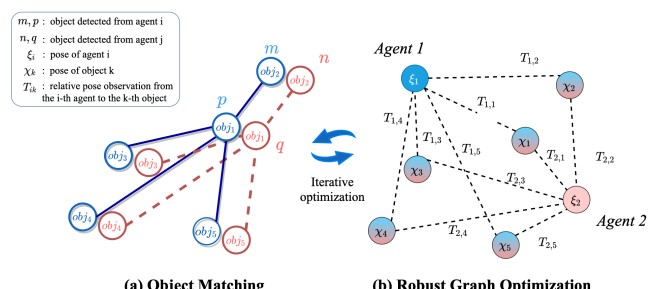

**(a) Object Matching**    **(b) Robust Graph Optimization**

**Figure 3: Object Matching and Pose graph illustration.**

## 4.1 Object Detection

Following the literature, the collaborative perception in this work uses LIDAR to perform 3D object detection. The input point cloud is in the dimensions of $(n \times 4)$, where $n$ denotes the number of points. Each point has its intensity and the 3D position in a world coordinate system. All agents use the same 3D object detector $\Phi_{Enc}$.

This work employs a standard uncertainty detection framework designed in the literature [25], which is based on the anchor-based PointPillar backbone [19]. Taking point clouds as input, this detection framework outputs features $F_i$ and the estimated bounding boxes for objects. In addition, it estimates the uncertainty of each bounding box, which is expressed as the variances of the center position and the heading angle of the corresponding object.

## 4.2 Graph-guided Object Matching

After performing object detection, agent $j$ will share a message $\mathcal{M}_{j \rightarrow i}$ to the ego agent $i$. This message contains three pieces of information including i) its feature map $F_j$; ii) the set of bounding boxes of the detect objects $\mathcal{B}_j$, in which each element has the form of $(x, y, z, w, l, h, \theta)$, denoting the 3D centre position, the size in different dimensions, and the yaw angle, respectively, for each bounding box; iii) the pose of agent $j$ denoted by $\xi_j = (x_j, y_j, \theta_j)$, representing its centre position and the heading angle on the 2D space. Once the ego agent $i$ receives the messages from all the other agents, it will start performing object matching by processing the messages one by one. Formally, the task can be described as follows: given the information $\{\mathcal{B}_i, \xi_i\}$ and $\{\mathcal{B}_j, \xi_j\}$, agent $i$ needs to find the associations among the bounding boxes in $\mathcal{B}_i$ and $\mathcal{B}_j$.

Modeling this task as a bipartite graph matching problem [14], we seek an optimal matching relationship that maximizes the similarity between the objects detected by agents $i$ and $j$. This can be formulated as

$$\mathcal{A}_{i,j}^* = \arg\max_{\mathcal{A}_{i,j}} \sum_{p \in \mathcal{B}_i} S\left(p, \mathcal{A}_{ij}(p)\right), \quad (2)$$

where $\mathcal{A}_{ij}$ denotes a list of one-to-one matching from the elements in $\mathcal{B}_i$ to those in $\mathcal{B}_j$. $\mathcal{A}_{ij}(p)$ represents the object, denoted by $q$, found in $\mathcal{B}_j$ by querying the matching relationship $\mathcal{A}_{i,j}$ using an object $p$ in $\mathcal{B}_i$. Let $S(p, q)$ denote the similarity of $p$ and $q$, where $q = \mathcal{A}_{ij}(p)$. A threshold $\tau_1$ is applied to select the reliable matching, i.e., $S(p, q) = 0$, if $S(p, q) < \tau_1$. Those selected ones will constitute $\mathcal{A}_{i,j}^*$, representing the obtained optimal matching relationship.

*4.2.1 Graph construction and Initial Matching.* The proposed object matching method is graph-guided. First, for each object $p$ in $\mathcal{B}_i$, we construct a star graph $\mathcal{G}_p$. Its central node is $p$ and the spatial neighbors of $p$ form other nodes of this graph, respectively. Each node has a feature vector consisting of the 3DoF pose $(x, y, \theta)$ of the object bounding box. In the same way, for each object $q$ in $\mathcal{B}_j$, a star graph $\mathcal{G}_q$ is constructed. They are illustrated in Figure 3(a).

To conduct initial matching, we transform $\mathcal{B}_j$ from agent $j$'s coordinate frame into agent $i$'s and establish the initial association using a distance-based method. For the two graphs $\mathcal{G}_p$ and $\mathcal{G}_q$, we evaluate the spatial distance between their central nodes $p$ and $q$, denoted by $dis(p, q)$. When the distance is shorter than a threshold $\tau_2$, $\mathcal{G}_p$ and $\mathcal{G}_q$ will be considered for matching. Note that multiple objects in $\mathcal{B}_j$ could have distances to $p$ shorter than $\tau_2$. The one with the minimum distance is chosen to be the initial match as

$$\mathcal{A}_{ij}(p) = \arg\min_{p \in \mathcal{B}_i, q \in \mathcal{B}_j} dis(p, q); \; where \; dis(p, q) \leq \tau_2 \quad (3)$$

*4.2.2 Graph Structure Similarity.* Logically, if $p$ and $q$ correspond to the same object, then the graphs $\mathcal{G}_p$ and $\mathcal{G}_q$ constructed from the two objects should have high similarity. Considering this, we design two types of similarity, edge similarity and distance similarity, to jointly measure the closeness of graphs $\mathcal{G}_p$ and $\mathcal{G}_q$ to assess whether $p$ and $q$ are from the same object.

**Edge Similarity.** In graph $\mathcal{G}_p$, the edge between the central node $p$ and one of its spatial neighbors $m$ is denoted by $e_{pm}$. It can be associated with a relative pose transformation from $p$ to $m$, denoted by $\mathbf{T}_{pm}$. $\mathbf{T}_{pm}$ is a $3 \times 3$ matrix obtained through the detection output of the objects $p$ and $m$. In the same way, in graph $\mathcal{G}_q$, the edges $e_{qn}$ between node $q$ and its neighbor $n$ can be associated with a relative pose transformation $\mathbf{T}_{qn}$. If $p$ and $q$ correspond to the same object, i.e., if $\mathcal{G}_p$ and $\mathcal{G}_q$ have a high similarity, then the edges in two graphs should be consistent. In this case, $\mathbf{T}_{qn}$ should be consistent with $\mathbf{T}_{pm}$. We define the following criterion to evaluate the edge consistency between $e_{pm}$ and $e_{qn}$:

$$l(e_{pm}, e_{qn}) = \exp(-\|\mathbf{T}_{pm}(\mathbf{T}_{qn})^{-1} - \mathbf{I}\|_F) \quad (4)$$

where $(\mathbf{T}_{qn})^{-1}$ is the inverse transformation of $e_{qn}$, $\mathbf{I}$ denotes the identity matrix, and $\| \cdot \|_F$ represents the Frobenius norm. The overall edge consistency score between $\mathcal{G}_p$ and $\mathcal{G}_q$ can be defined as:

$$S_{edge}(p, q) = \frac{1}{|N_p|} \sum_{m \in N_p} l(e_{pm}, e_{qn}), \quad s.t. \quad n = \mathcal{A}_{ij}(m) \quad (5)$$

where $N_p$ contains the leaf vertices in $\mathcal{G}_p$ that have corresponding vertices in $\mathcal{G}_q$. Note that the object $n$ is the one found to correspond to object $m$ in the list $\mathcal{A}_{ij}$.

**Distance Similarity.** In addition to edge similarity, we can also further quantify the closeness of the two graphs by using distance similarity. This is because if the central nodes $p$ and $q$ correspond to the same object, the spatial distance between the detection boxes should be sufficiently close, even in the presence of noise and or in a crowded situation. We define the distance similarity of the graphs of $p$ and $q$ as:

$$S_{dis}(p, q) = \exp(-dis(p, q)) \quad (6)$$

where $dis(\cdot, \cdot)$ is implemented as a Euclidean distance between the coordinates of the centers of the corresponding bounding boxes.

Therefore, the overall graph similarity is obtained by combining the edge consistency and the distance consistency as

$$S(p,q) = S_{edge}(q,p) + \lambda S_{dis}(p,q) \tag{7}$$

where $\lambda$ is the hyperparameter for balancing.

Finally, we utilize the overall similarity $S$ to implement the maximization problem in Equation (7) to find the optimal matching. Essentially, this is a bipartite graph problem, where one graph corresponds to all bounding boxes $p$ detected by agent $i$, and the other corresponds to all bounding boxes $q$ detected by agent $j$. The previously defined similarity $S(p,q)$, serves as the edge weight in the bipartite graph. Solving this maximum matching problem can be accomplished using the Kuhn-Munkres algorithm [48].

## 4.3 Robust Pose Graph Optimization

After performing object matching, the elements in the pair of object detection box sets $\mathcal{B}_i$ and $\mathcal{B}_j$ will establish one-to-one correspondences. As shown in Figure 3(b), we then construct a pose graph $\mathcal{G}\left(\mathcal{V}_{agent}, \mathcal{V}_{object}, \mathcal{E}\right)$ for all the agents and the detected objects in the whole scene and use this correspondence to adjust their poses. The node set $\mathcal{V}_{agent}$ comprises all $N$ agents, while the node set $\mathcal{V}_{object}$ consists of all detected objects, and the edge set $\mathcal{E}$ represents the detection relationships between agents and objects. As mentioned at the beginning of Section 4, there could still be incorrect matching relationships between objects, even after the process proposed in Section 4.2, and this will certainly affect the adjustment of the poses of agents. Therefore, we develop an iterative process as what follows in order to achieve robust object matching and pose adjustment.

Firstly, after obtaining the object matching relationship $\mathcal{A}_{i,j}$, as in Section 4.2, we will perform pose adjustment guided by this matching relationship; After this, the elements of $\mathcal{B}_i$ and $\mathcal{B}_j$ will be updated; We then use the updated $\mathcal{B}_i$ and $\mathcal{B}_j$ to perform a new round of object matching as in Section 4.2, and then a new round of pose adjustment will be conducted. This iterative matching and adjustment process repeats until the matching result converges, yielding the optimal matching result $\mathcal{A}_{i,j}^*$ and the optimized poses $\xi_j$ and $\chi_k$. The process is illustrated in Figure 3(a)(b). We model this task as a graph optimization problem. In this pose graph, each node is associated with a pose. The pose of the $j$th agent is denoted as $\xi_j$, and the pose of the $k$th object is denoted as $\chi_k$. Let us assume that after the step in Section 4.2, $p$ and $q$ form a matching pair $(\mathcal{A}_{ij}(p) = q)$ and they are the bounding boxes of the same object.

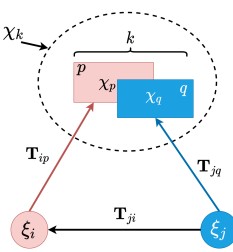

**Figure 4: How the residue errors are set up.**

Let $\mathbf{T}_{ip}$ be the measured relative transformation of the $p$th object from the perspective of the $i$th agent, which is naturally obtained through the $i$th agent's detection output; $\mathbf{T}_{jq}$ is the measured relative transformation of the $q$th object from the perspective of the $j$th agent. From agent $i$'s observation, we have $\mathbf{T}_{ip} = \mathbf{E}_i^{-1}\mathbf{X}_p$, so $\mathbf{X}_p = \mathbf{E}_i\mathbf{T}_{ip}$. Note that $\mathbf{E}_i$ is a matrix constructed from $\xi_i$ which denotes the pose of agent i [1]. Similarly, we can get $\mathbf{X}_q = \mathbf{E}_j\mathbf{T}_{jq}$. In the optimization process, the variables to optimize in this scenario are $\{\xi_j, \chi_k\}$. The matched objects are hoped to converge to a consistent result $\chi_p = \chi_q = \chi_k$. An example is given in Figure 4 to show how the residue errors are set up. We can set up the residue error functions:

$$e_{ik} = (\mathbf{E}_i\mathbf{T}_{ip})^{-1}\mathbf{X}_k = \mathbf{T}_{ip}^{-1}\mathbf{E}_i^{-1}\mathbf{X}_k \tag{8}$$

$$e_{jk} = (\mathbf{E}_j\mathbf{T}_{jq})^{-1}\mathbf{X}_k = \mathbf{T}_{jq}^{-1}\mathbf{E}_j^{-1}\mathbf{X}_k \tag{9}$$

$$e_{ji} = \mathbf{T}_{ji}^{-1}\mathbf{E}_j^{-1}\mathbf{E}_i \tag{10}$$

The overall optimization objective can be given as:

$$\left\{\xi_j', \chi_k'\right\} = \underset{\{\xi_j, \chi_k\}}{\arg\min} \sum_{(j,k)} \left(e_{ik}^T\Omega_{ik}e_{ik} + e_{jk}^T\Omega_{jk}e_{ik} + e_{ij}^T\Omega_{ij}e_{ij}\right) \tag{11}$$

Where $\Omega = \text{diag}\left(\sigma_x^{-2}, \sigma_y^{-2}, \sigma_\theta^{-2}\right) \in \mathbb{R}^{3\times3}$ is the information matrix, and its diagonal elements come from the uncertainty estimates of the bounding box, which are part of the detection output from agent $j$. By solving (11) using Levenberg-Marquardt algorithm [20], we get $\{\xi_j', j = 1 : N\}$, $\{\chi_k', k = 1 : K\}$. Then we re-do the object matching result:

$$\mathcal{A}_{ij}' = \Phi_{Match}\left(\left\{\mathcal{B}_i, \xi_i\right\}, \left\{\mathcal{B}_j, \xi_j'\right\}\right) \tag{12}$$

In the iterative matching (12) and pose optimization (11) process, we update the position of $\mathcal{B}_j$ using $\xi_j'$. our goal is to gradually improve object matching results by minimizing the overall graph optimization error. During the iterative matching process, we fix the poses of self-agent and update the poses of other agents and the objects.

## 4.4 Aggregation and Detection

After the pose calibration, the corrected relative pose from agent $j$ to agent $i$ is denoted as $\xi_{j\rightarrow i}' = \xi_i^{-1} \circ \xi_j'$. With the corrected relative pose $\xi_{j\rightarrow i}'$, we can adjust the contents of $\mathcal{M}_{j\rightarrow i}$, feature $\mathbf{F}_j$ from agent $j$ are synchronized to the ego pose which has the same coordinate system with its ego feature $\mathbf{F}_i$. After the spatial alignment, each agent aggregates $\Phi_{Agg}(\cdot)$ other agents' collaboration information and obtain a more informative feature. This aggregating function can be any common fusion operation. All our experiments adopt Multi-scale feature fusion. After receiving the final fused feature maps, we decode them into the final detection layer $\Phi_{Dec}(\cdot)$ to obtain final detections $O_i$. The regression output

---

[1]Given the vector $\xi = (x, y, \theta)$, it can be converted into a transformation matrix:
$\mathbf{E} = \begin{bmatrix} \cos(\theta) & -\sin(\theta) & x \\ \sin(\theta) & \cos(\theta) & y \\ 0 & 0 & 1 \end{bmatrix}$, and the conversion between $\mathbf{X}$ and $\chi$ follows the same way. Recall that $\chi$ denotes the pose of an object.

is $(x, y, z, w, l, h, \theta)$, denoting the position, size, yaw angle of the anchor boxes, respectively. The classification output is the confidence score of being an vehicle or background for each anchor box. The overall process of RoCo is given in Algorithm 1.

---

**Algorithm 1:** Collaborative Perception with RoCo

---

1 **Input data:** raw observation: $\mathcal{X}_i$
2 $F_i, \mathcal{B}_i = \Phi_{Enc}(\mathcal{X}_i), \xi_{j \to i} = \xi_j$
3 **Sharing messages** $\{F_j, \mathcal{B}_j, \xi_j\}$ **to** $i$:
4 **Initialize** $\mathcal{A}_{ij}$ using Eq.(3)
5 **while** *Not Converged* **do**
6     $\left\{\xi'_j\right\} = \Phi_{Rbst}\left(\mathcal{A}_{ij}, \left\{\mathcal{B}_j, \xi_j\right\}_{j=1,2,..,N}\right)$
7     $\mathcal{A}_{ij} = \Phi_{Match}\left(\left\{\mathcal{B}_i, \xi_i\right\}, \left\{\mathcal{B}_j, \xi'_j\right\}\right),$ $\quad j=1,\cdots,N; j\neq i$
8 **end**
9 $F'_j = \Phi_{Proj}\left(\left(F_j, \xi'_{j \to i}\right)\right)$
10 $F'_i = \Phi_{Agg}\left(F_i, \left\{F'_j\right\}_{j=1,2,...N, j\neq i}\right)$
11 $O_i = \Phi_{Dec}\left(F'_i\right)$

---

## 5 EXPERIMENTAL RESULT

We conduct extensive experiments on both simulated and real-world scenarios. The task is point-cloud-based 3D object detection. Following the literature, the detection results are evaluated by Average Precision (AP) at Intersection-over-Union (IoU) threshold of 0.50 and 0.70.

### 5.1 Dataset

**DAIR-V2X [44].** DAIR-V2X is a public real-world collaborative perception dataset. It contains two agents: vehicle and road-side-unit (RSU) with image resolution $1080 \times 1920$. The perception range is $x \in [-100m, 100m], y \in [-40m, 40m]$. RSU LiDAR is 300-channel while the vehicle's is 40-channel. **V2XSet [40].** V2XSet is a large-scale dataset designed for Vehicle-to-Infrastructure (V2X) communication. The data in V2XSet is collected using the simulator CARLA [10] and OpenCDA [28]. It includes LiDAR data captured from multiple autonomous vehicles and roadside infrastructure in various scenarios. The dataset consists of a total of 11,447 frames, with train, validation, test splits of 6,694/1,920/2,833 frames, respectively.

### 5.2 Implementation Details

For encoder backbone, we use PointPillar [19] with the voxel resolution to 0.4m for both height and width. To simulate pose errors, we follow the noisy settings in CoAlign [25] during the training process. We add Gaussian noise $N\left(0, \sigma_t^2\right)$ on $x, y$ and $N\left(0, \sigma_r^2\right)$ on $\theta$, where $x, y, \theta$ are the coordinates of the 2D centers of a vechicle and the yaw angle of accurate global poses. We use the bounding boxes output by the detection framework in reference[25] for each individual agent. In the bipartite graph matching, we set $\lambda = 1$ for Eq.(7) and similarity threshold $\tau_1 = 0.5$ for Eq.(2). In the pose graph optimization, the Levenberg-Marquardt algorithm[20] is employed to solve the least squares optimization problem, with the maximum

number of iterations set to 1000. To better contrast the performance of RoCo, we retrained other methods. We use the Adam [17] with an initial learning rate 0.001 for detection and 0.002 for feature fusion. The batchsizes we set for the DAIR-V2X and V2XSet datasets are 4 and 2, respectively. All models are trained on six NVIDIA RTX 2080Ti GPUs with epoch number 30.

### 5.3 Quantitative evaluation

To validate the overall performance of RoCo in 3D object detection, we compare it with a series of previous methods on two datasets. For a fair comparison, all models take 3D point clouds as input data, RoCo is compared with seven state-of-the-art methods: F-Cooper [8], FPV-RCNN[46], V2VNet [35], V2X-ViT[40], Self-ATT, CoAlign [25] and CoBEVFlow[36]. Table 1 shows the AP at IoU threshold of 0.5 and 0.7 in DAIR-V2X and V2XSet dataset. We reference the results from the literatures [25, 36] in the DAIR-V2X and implement them in the V2XSet, and we also implement noise level of $\sigma_t^2/\sigma_r^2 = 0.8m/0.8°$ on both datasets. We see that RoCo significantly outperforms the previous methods at various noise levels across both datasets. In the real-world dataset DAIR-V2X, RoCo outperforms CoAlign across all noise settings. In the case of noise levels of $0.0m/0.0°$, our approach achieves 1.7% (76.3% *vs.* 74.6%) and 1.6% (62.0% *vs.* 60.4%) improvement over CoAlign for AP@0.5/0.7. When the noise level becomes as high as $0.6m/0.6°$, our approach achieves 1.9% (71.9% *vs.* 70.20%) and 1.2% (58.2% *vs.* 57.0%) improvement over CoAlign for AP@0.5/0.7. In the case of noise levels of 0.8/0.8, our approach still achieves 2.3% (71.5% *vs.* 69.2%) and 0.9% (57.6% *vs.* 56.9%) improvement over CoAlign for AP@0.5/0.7. We conduct experiments on a simulated dataset V2XSet which involves more agents. RoCo still performs well across all noise settings, as shown in Table 1. As the level of noise increases, both methods experience performance degradation. However, our method maintains a higher level of accuracy even under high-level of noise. In the case of noise level of $0.4m/0.4°$, our approach achieves 1.9% (90.0% *vs.* 88.1%) and 4.3% (77.3% *vs.* 73.0%) improvement over CoAlign for AP@0.5/0.7. When the noise level reaches $0.8m/0.8°$, our approach still achieves 1.4% (84.1% *vs.* 82.7%) and 1.6% (68.9% *vs.* 67.3%) improvement over CoAlign for AP@0.5/0.7.

### 5.4 Qualitative evaluation

Figure 5 and Figure 6 show the 3D detection results in the BEV format on the V2XSet. Red and green boxes denote the detection results and the ground-truth, respectively. The degree of overlapping of these boxes reflects the performance of the testing methods.

Figure 5 depicts the detection results of V2X-ViT, V2VNet, CoAlign, and the proposed RoCo at an intersection to validate the effectiveness of our method. We set the noise level of $0.4m/0.4°$ and $0.8m/0.8°$ to produce high pose errors to make the perception task challenging. From the figure, it can be observed that V2VNet has many missed detections, while V2X-ViT and CoAlign generate many predictions with relatively large offsets. In contrast, our RoCo demonstrates strong performance high pose errors. Figure 6 visualize the box position with and without graph matching in crowded scenes. Without the proposed graph matching, show in figure 6(a), there are many false detections. In comparison, each detection box in Figure 6(b) corresponds to a unique object, exhibiting

**Table 1: 3D object detection performance on DAIR-V2X[44] and V2XSet[40] datasets. All models are trained on pose noises following $\sigma_t = 0.2m, \ \sigma_r = 0.2°$. Experiments show that RoCo achieves the best performance under various noise levels.**

| Dataset | DAIR-V2X | | | | | V2XSet | | | | |
|---|---|---|---|---|---|---|---|---|---|---|
| Method/Metric | AP@0.5 ↑ | | | | | | | | | |
| Noise Level $\sigma_t/\sigma_r$ ($m/°$)) | 0.0/0.0 | 0.2/0.2 | 0.4/0.4 | 0.6/0.6 | 0.8/0.8 | 0.0/0.0 | 0.2/0.2 | 0.4/0.4 | 0.6/0.6 | 0.8/0.8 |
| F-Cooper[8] | 73.4 | 72.3 | 70.5 | 69.2 | 67.1 | 78.3 | 76.3 | 71.2 | 65.9 | 62.0 |
| FPV-RCNN[46] | 65.5 | 63.1 | 58.0 | 58.1 | 57.5 | 86.5 | 85.3 | 68.7 | 62.1 | 49.5 |
| V2VNet[35] | 66.0 | 65.5 | 64.6 | 63.6 | 61.7 | 87.1 | 86.0 | 83.2 | 79.7 | 75.0 |
| Self-Att[42] | 70.5 | 70.3 | 69.5 | 68.5 | 67.8 | 87.6 | 86.8 | 85.4 | 83.7 | 82.1 |
| V2X-ViT[40] | 70.4 | 70.0 | 68.9 | 67.8 | 66.0 | 91.0 | 90.1 | 86.9 | 84.0 | 81.8 |
| CoAlign[25] | 74.6 | 73.8 | 72.0 | 70.0 | 69.2 | 91.9 | 90.9 | 88.1 | 85.5 | 82.7 |
| CoBEVFlow[36] | 73.8 | 73.2 | 70.3 | - | - | - | - | - | - | - |
| Ours (RoCo) | 76.3 | 74.8 | 73.3 | 71.9 | 71.5 | 91.9 | 91.0 | 90.0 | 85.9 | 84.1 |

| Method/Metric | AP@0.7 ↑ | | | | | | | | | |
|---|---|---|---|---|---|---|---|---|---|---|
| Noise Level $\sigma_t/\sigma_r$ ($m/°$)) | 0.0/0.0 | 0.2/0.2 | 0.4/0.4 | 0.6/0.6 | 0.8/0.8 | 0.0/0.0 | 0.2/0.2 | 0.4/0.4 | 0.6/0.6 | 0.8/0.8 |
| F-Cooper[8] | 55.9 | 55.2 | 54.2 | 53.8 | 51.6 | 48.6 | 46.0 | 43.4 | 41.0 | 39.5 |
| FPV-RCNN[46] | 50.5 | 45.9 | 42.0 | 41.0 | 38.9 | 56.3 | 51.2 | 37.4 | 31.8 | 27.0 |
| V2VNet[35] | 48.6 | 48.3 | 47.8 | 47.5 | 38.0 | 64.6 | 62.0 | 56.2 | 50.7 | 44.9 |
| Self-Att[42] | 52.2 | 52.0 | 51.7 | 51.4 | 51.1 | 67.6 | 66.2 | 65.1 | 63.9 | 63.0 |
| V2X-ViT[40] | 53.1 | 52.9 | 52.5 | 52.2 | 51.3 | 80.3 | 76.8 | 71.8 | 69.0 | 66.6 |
| CoAlign[25] | 60.4 | 58.8 | 57.9 | 57.0 | 56.9 | 80.5 | 77.3 | 73.0 | 70.1 | 67.3 |
| CoBEVFlow[36] | 59.9 | 57.9 | 56.0 | - | - | - | - | - | - | - |
| Ours (RoCo) | 62.0 | 59.4 | 58.4 | 58.2 | 57.8 | 80.5 | 77.4 | 77.3 | 71.0 | 68.9 |

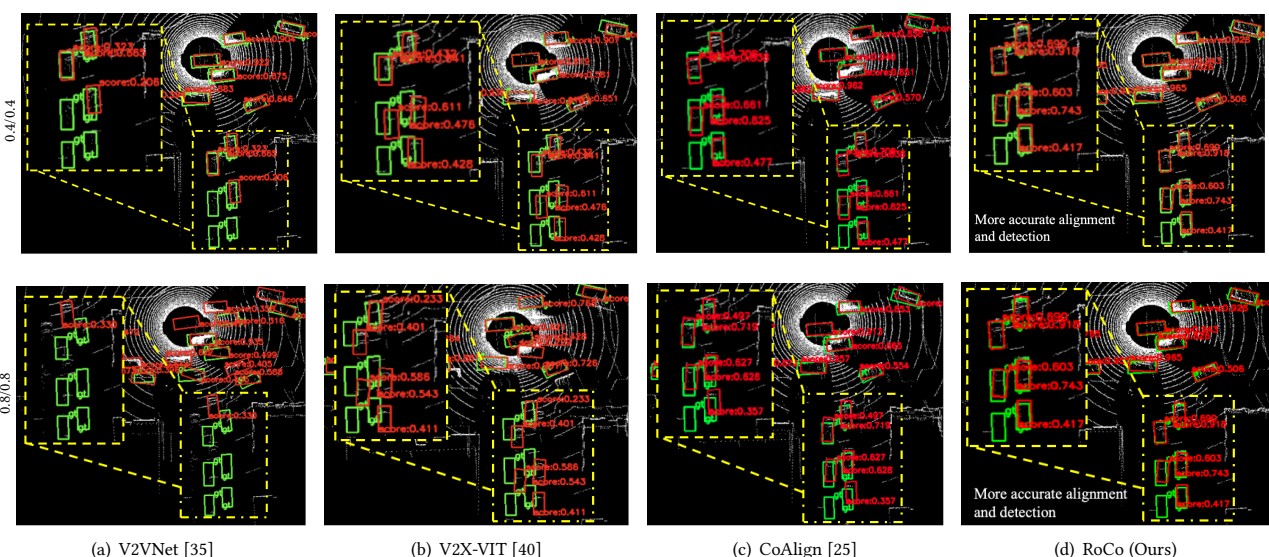

(a) V2VNet [35]      (b) V2X-VIT [40]      (c) CoAlign [25]      (d) RoCo (Ours)

**Figure 5: Visualization of detection results for V2VNet, V2X-ViT, CoAlign and our RoCo with the noisy level $\sigma_t^2/\sigma_r^2$ ($m/°$) of 0.4/0.4 (the first row), and 0.8/0.8 (the second row) on V2XSet dateset. An intersection scenario is given. RoCo qualitatively outperforms the others under different noisy level.**

more robust performance. In crowded environments, objects are relatively close and with high pose errors, the detected bounding boxes may overlap. RoCo utilizes graph-based object matching to correctly associate objects detected by different agents, ensuring that each object has only one correct detection box.

## 5.5 Ablation Studies

**Selection of the threshold value $\tau_2$.** The selection of the initial candidate set in graph matching is crucial. To determine the optimal threshold $\tau_2$ in Eq.(3) during the graph initialization, we conduct ablation experiments on different datasets, as shown in Table 2. In

**Table 2: Selection of the threshold value $\tau_2$ in Eq. (3).**

| Threshold/Metric | DAIR-V2X | | | | V2XSet | | | |
|---|---|---|---|---|---|---|---|---|
| | AP@0.7 | | | | | | | |
| Noise Level | 0.2/0.2 | 0.4/0.4 | 0.6/0.6 | 0.8/0.8 | 0.2/0.2 | 0.4/0.4 | 0.6/0.6 | 0.8/0.8 |
| $\tau_2 = 1$ | 58.8 | 57.6 | 56.9 | 56.4 | 77.1 | 72.5 | 70.4 | 65.7 |
| $\tau_2 = 2$ | 58.9 | 57.9 | 57.2 | 56.9 | **77.4** | **77.3** | **71.0** | 68.1 |
| $\tau_2 = 3$ | **59.4** | **58.4** | **58.2** | **57.8** | 77.1 | 73.1 | 70.2 | **68.9** |
| $\tau_2 = 4$ | 57.9 | 57.6 | 57.6 | 57.5 | 75.9 | 72.0 | 69.7 | 68.3 |

**Table 3: Contribution of graph similarity.**

| Modules | | | AP@0.5 | | | AP@0.7 | | |
|---|---|---|---|---|---|---|---|---|
| Matching | distance | edge | 0.2/0.2 | 0.4/0.4 | 0.8/0.8 | 0.2/0.2 | 0.4/0.4 | 0.8/0.8 |
| ✗ | ✗ | ✗ | 73.8 | 72.0 | 69.2 | 58.8 | 57.9 | 56.9 |
| ✓ | ✓ | ✗ | 73.0 | 71.0 | 70.8 | 58.8 | 58.0 | 57.5 |
| ✓ | ✗ | ✓ | 73.3 | 72.1 | 70.9 | 59.0 | 58.1 | 57.6 |
| ✓ | ✓ | ✓ | **74.8** | **73.3** | **71.5** | **59.4** | **58.4** | **57.8** |

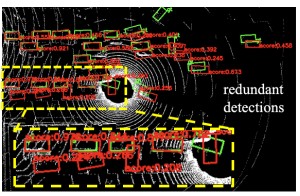 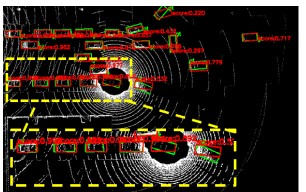

(a) Boxes w/o Graph matching      (b) Boxes w/ Graph matching

**Figure 6: 3D object detection results in congested traffic scenario.**

**Table 4: Ablation study on DIAR-V2X.**

| Modules | | AP@0.5 | | | AP@0.7 | | |
|---|---|---|---|---|---|---|---|
| Matching | Robust Optimization | 0.2/0.2 | 0.4/0.4 | 0.8/0.8 | 0.2/0.2 | 0.4/0.4 | 0.8/0.8 |
| ✗ | ✗ | 73.8 | 72.0 | 69.2 | 58.8 | 57.9 | 56.9 |
| ✓ | ✗ | 74.4 | 72.9 | 71.0 | 59.1 | 58.2 | 57.5 |
| ✓ | ✓ | **74.8** | **73.3** | **71.5** | **59.4** | **58.4** | **57.8** |

the DAIR-V2X dataset, we found that the detection results reach their optimum when $\tau_2$ is set to 3. As the value of $\tau_2$ increases, the detection performance will decline. This is because vehicles usually maintain a safe distance from other vehicles in real-world scenarios, and too small a threshold would affect the initial matching. Furthermore, we conducted the same experiments on the simulated dataset. We observed that when the noise level is below $0.8m/0.8°$, the optimal threshold $\tau_2$ becomes 2 meters. As the noise level increases to $0.8m/0.8°$, the optimal threshold becomes $\tau_2 = 3$. This is because in the V2XSet dataset, there are more vehicles and they are densely distributed. As a result, the distances between vehicles are smaller compared to real-world scenarios.

**Contribution of Object Matching and Robust Graph Optimization.** Table 4 assesses the effectiveness of the proposed module at various noise levels, including object matching and robust graph optimization. We evaluate the impact of each module on the final 3D detection by incrementally adding i) object matching and ii) robust graph optimization. We see that both modules can improve

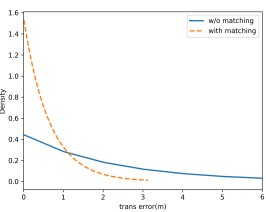 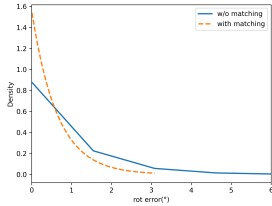

**Figure 7: Probability density function (PDF) of relative pose error distribution on V2XSet dataset when noise level is $0.8m/0.8°$.**

the performance. To validate the impact of object matching on pose error, Figure 7 plots the distribution of relative pose errors across all samples. The horizontal axes "trans error" and "rot error" denote the translational and rotational relative pose errors (RPE), respectively. The vertical axis represents the density value of the distribution. The orange dashed line represents the density when applying the proposed object matching method, while the blue line represents case without the object matching. The closer the distribution is to the Delta distribution centered at zero, the better the performance because this means most of the pose errors have smaller values. It can be observed from figure that using object matching leads to the desired distribution and significantly reduces relative pose errors.

**Contribution of graph similarity.** Now we investigate the contributions of different similarities defined in Eq.(5) and (6) to object matching on DARI-V2X, and the result is shown in Table 3. We evaluate the impact of each similarity score on the final graph matching by incrementally adding i) edge similarity and ii) distance similarity. The first row in Table 3 represents the baseline model which does not use any matching methods. The experimental results indicate that both designs of similarity matching contribute to improving detection accuracy, especially at the noise level of $0.8m/0.8°$. The proposed method outperforms the baseline model by 2.3% (71.5% *vs.* 69.2%) and achieves an AP@0.5/0.7 improvement of 0.9% (57.8% *vs.* 56.9%).

## 6 CONCLUSION

This paper proposes a novel robust collaborative perception framework, **RoCo**, for 3D object detection. This framework addresses issues such as object mismatch and misalignment caused by pose errors among multiple agents. It enhances the precision and reliability of this modality and better prepare it for the potential multimodal fusion step in the sequel. The core idea of **RoCo** is based on graph-based object matching, which reliably associates common objects detected by different agents and reduces pose errors of agents and objects using iterative robust optimization. Additionally, **RoCo** does not require any ground truth pose supervision during training, making it highly practical. Comprehensive experiments demonstrate that **RoCo** achieves outstanding performance across all settings and exhibits superior robustness under extreme noise conditions. We believe that this work has the potential to contribute to the development of the multimodal field. In future work, we will apply our method to multimodal collaborative perception to further improve the safety and reliability of autonomous driving.

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
