# OpenReview forum: "RoCo: Robust Cooperative Perception By Iterative Object Matching and Pose Adjustment"
_acmmm.org/ACMMM/2024/Conference — MM2024 Oral_

### Official Review · Reviewer_7rAw · 2024-05-16

**Rating:** 6
**Confidence:** 3

**Summary:**

The study addresses the issue in collaborative perception where the quality of modality-based object detection is highly sensitive to the relative pose errors between agents, leading to feature inconsistency and reduced collaborative performance. An unsupervised learning method is proposed, modeling the pose correction problem as an object matching task, and adjusting agent poses through a graph optimization process to minimize the alignment errors of associated objects. Ultimately, the method demonstrates superiority over existing relevant methods on both simulated and real-world datasets and exhibits high robustness in noisy environments.

**Strengths:**

1. By constructing an optimization graph, RoCo is able to iteratively adjust agent poses based on global observational consistency, effectively filtering out incorrect matches.
2. Even in scenarios with vehicle congestion and high noise, RoCo can accurately align the detection results of the same object obtained by different agents, showing good robustness.
3. RoCo does not require any ground truth during training and performs well on both simulated and real-world datasets, making it highly practical.
4. Experiments conducted on the DAIR-V2X and V2XSet datasets demonstrate that this method has achieved performance improvements compared to other methods.

**Limitations:**

1. The paper considers a scenario where each agent can accurately detect objects within its own range. In practical scenarios, if the initial pose estimation is inaccurate, the optimization process may require more iterations to converge, resulting in a greater computational load during training, which needs to be considered in future work.
2. In some resource-constrained scenarios, it is uncertain whether the trained RoCo model maintains an advantage in speed compared to other solutions.
3. The paper mentions several key parameters, such as the similarity threshold and the initial candidate set for graph matching, the selection of these parameters has a certain impact on the system performance. The psper may need to further discuss how these parameters should be set.
4. The RoCo framework proposed in the article involves graph optimization and iterative matching processes, which may lead to high computational complexity in scenarios with a large number of agents and objects.

**Suitability:**

3

---

### Official Review · Reviewer_oqJv · 2024-05-17

**Rating:** 5
**Confidence:** 3

**Summary:**

This paper proposes a new iterative object matching and pose adjustment object detection framework (Roco), which can handle the matching problem of multi-vehicle object detection information in noisy environments. Roco is divided into object matching and robust graph optimization. Object matching uses edge similarity and distance similarity in star graphs to accurately establish matching relationships between multiple detection objects. Robust graph optimization constructs a pose optimization graph for the entire scene based on the above matching relationships, and iteratively adjusts the agent's pose based on global consistency, effectively filtering out incorrect matches.

**Strengths:**

1. This is an unsupervised method that does not require ground-truth pose information for the agents or objects and it can adapt to various levels of pose errors.
2. Implementing RoCo does not require retraining or fine-tuning any network models and can be integrated into any 3D object detection-based collaborative perception framework.
3. Even in crowded and noisy environments, RoCo can accurately align the detections of the same object obtained by different agents.

**Limitations:**

1.What is the difference between the proposed star graph and the star map of ‘Human–Object Interaction Detection Based on Star Graph’.

2.The citation format of some references is incorrect, and there is an inconsistency between uppercase and lowercase.

**Suitability:**

2

---

### Official Review · Reviewer_YLgS · 2024-05-19

**Rating:** 4
**Confidence:** 3

**Summary:**

The paper pays attention on the pose correction problem in crowded and complex scenarios with a iterative object matching and pose adjustment framework called RoCo. The paper models the pose correction problem in collaborative perception as an object matching task, and proposes graph-guided object matching and pose graph optimization to improve 3D object detection performance. Experimental results show that RoCo achieves more accurate and robust 3D object detection performance

**Strengths:**

1.A multi-agent collaborative LiDAR-based 3D object detection framework that addresses the matching errors and pose inaccuracies between agents and objects.

2.A graph matching approach that establishes matching relationships between targets based on distance and neighborhood structural consistency.

3.An optimization method that iteratively adjusts agent poses based on global observation consistency.

4.Experimental results show that RoCo achieves more accurate and robust 3D object detection performance.

**Limitations:**

1 The technical contributions should be expressed more clearly. In the paper, the authors models the pose correction problem of multiple agents as an object matching and optimization task. To solve the problem, the paper proposes graph based object matching and optimization while stating the detailed process. However, the novelties are not well addressed. Compared with other object matching methods, what are the novelties of the proposed graph based object matching and optimization? Moreover, some SLAM systems also employ graph optimization to refine camera pose and feature points. Compared with this, what is main contribution of the proposed graph optimization?

2 In collaborative perception, the inference time is also an important factor. The experiments of running time analysis are missing. It is suggested to add some comparisons about running time.

3 Some minors.

(1)Some reference citations are not consistent with others, like Line 172, Line 230.

(2)Line 327, “To improve this situation”, it is suggested to modify it to clearly express the situation.

(3)It seems that there is some problems about the supplementary video. Please check it.

**Suitability:**

2

---

### Official Review · Reviewer_KnMo · 2024-05-23

**Rating:** 4
**Confidence:** 4

**Summary:**

This paper introduces a novel unsupervised framework for iterative object matching and agent pose adjustment. The framework models the pose correction problem in collaborative perception as an object matching task, reliably associating common objects detected by different agents. Concurrently, it employs a graph optimization process to adjust the agent poses by minimizing the alignment errors of the associated objects. The object matching is then re-executed based on the adjusted agent poses. This iterative process continues until convergence is achieved. Experimental studies on both simulated and real-world datasets demonstrate that the proposed framework, RoCo, consistently outperforms existing methods in terms of collaborative object detection performance. Additionally, RoCo exhibits robust performance even when the pose information of agents is highly noisy. Ablation studies further illustrate the impact of key parameters and components on the framework's performance.

**Strengths:**

1. The model design of this paper is well-motivated. The authors have devised a framework to address a major challenge in cooperative perception: localization error.
2. The paper demonstrates significant performance improvements on the two most popular datasets, DAIR-V2X and V2Xset.
3. Overall, this paper is well-written.

**Limitations:**

1. Although the abstract claims that this paper is the first to model the pose correction problem as an object matching task, this idea has already been proposed in Feaco[1]. The paper should include a comparison with Feaco in the experiments.

2. Despite the proposed model showing prominent accuracy compared to other state-of-the-art methods, the model appears complex. What is its real-time performance?

3. Why were no experiments conducted on OPV2V? The experiment codes for OPV2V and V2Xset do not require any adjustments. I'm curious about the results on OPV2V.

The authors' work is well-motivated and novel, but some concerns need to be addressed in the rebuttal. I will change my rating based on the rebuttal.

[1] FeaCo: Reaching Robust Feature-Level Consensus in Noisy Pose Conditions , MM2023

**Suitability:**

3

---

### Meta-Review · Area_Chair_AiCC · 2024-06-26

**Recommendation:** Accept (Oral)
**Confidence:** 5

**Metareview:**

This paper introduces RoCo, an unsupervised framework for iterative object matching and agent pose adjustment in collaborative perception scenarios. The proposed method models pose correction as an object matching task, employing a graph optimization process to minimize alignment errors. Experimental results on DAIR-V2X and V2Xset datasets demonstrate significant performance improvements over existing methods, showcasing RoCo's robustness even in highly noisy environments.

The paper is well-written, and the authors have addressed most concerns raised by the reviewers. The novelty of modeling pose correction as an object matching task, combined with robust graph optimization, offers a valuable contribution to the field. The main limitations include the complexity of the model and the lack of experiments on OPV2V, which should be addressed in future work.